# Plastic Covers and Potassium Pre-Harvest Sprays and Their Influence on Antioxidant Properties, Phenolic Profile, and Organic Acids Composition of Sweet Cherry Fruits Cultivated in Southern Chile

**DOI:** 10.3390/plants12010050

**Published:** 2022-12-22

**Authors:** Cristóbal Palacios-Peralta, Antonieta Ruiz, Stefano Ercoli, Marjorie Reyes-Díaz, Marco Bustamante, Ariel Muñoz, Pamela Osorio, Alejandra Ribera-Fonseca

**Affiliations:** 1Centro de Fruticultura, Facultad de Ciencias Agropecuarias y Forestales, Campus Andrés Bello, Universidad de La Frontera, Avenida Francisco Salazar, P.O. Box 54-D, Temuco 01145, Chile; 2Departamento de Ciencias Químicas y Recursos Naturales, Facultad de Ingeniería y Ciencias, Campus Andrés Bello, Universidad de La Frontera, Avenida Francisco Salazar, P.O. Box 54-D, Temuco 01145, Chile; 3Center of Plant-Soil Interaction and Natural Resources Biotechnology, Scientific and Technological, Bioresource Nucleus (BIOREN), Campus Andrés Bello, Universidad de La Frontera, Avenida Francisco Salazar, P.O. Box 54-D, Temuco 01145, Chile; 4Research, Development and Innovation Department, Exportadora Rancagua S.A.—Ranco Cherries, Route 5 South, Km 80, P.O. Box 576, Rancagua 04000, Chile

**Keywords:** total phenolic content, flavonoids anthocyanins, fruit quality, macro nutrient

## Abstract

In rainy areas, sweet cherries are cultivated under plastic covers, preventing the cracking of the fruit but decreasing the firmness and acidity of the cherries. We evaluated the impact of plastic cover and pre-harvest K foliar application on quality parameters, antioxidant properties, and phenolic and organic acid compositions in fruits of sweet cherry cv. Regina of Southern Chile. Our results showed that K+ increased firmness, total soluble content, size, fruit weight, and titratable acidity at harvest, independent of the cover factor. The positive impacts of foliar K fertilization on anthocyanins, flavonoids, and phenolic acids could explain the higher antioxidant capacity of fruits. Our study revealed that the additional K doses applied increased malic acid, the main organic acid in cherry fruits, but only in fruits from uncovered trees. In covered trees, the effect was reversed. Citric acid was higher in fruit from covered trees. Our results indicated that tartaric acid also increased with the application of higher K doses; however, this acid was detectable only in uncovered tree fruit. Interestingly, all organic acids were lower in fruits produced in the lower canopy than those detected in fruits harvested from the upper canopy. This showed the positive impact of canopy light exposure on maintaining suitable acidity levels in sweet cherry fruits.

## 1. Introduction

Sweet cherry (*Prunus avium* L.) is one of the most important fruit crops cultivated in temperate climates [1]. Currently, Chile reached third place among countries with the highest sweet cherry production worldwide, after Turkey and the United States of America [2]. It is noteworthy that this country boosted sweet cherry production from 41,000 to 234,000 tons in the last 10 years. According to a recent report of the USDA, sweet cherry area planted in Chile reached a total of 49,000 ha in the 2021/22 season, which are mainly located in the Central Zone [3]. It was highlighted that, in the last years, plantation area is expanding to southern Chile from La Araucanía to Los Lagos Regions [4], which is currently considered a very attractive zone for this fruit tree due to the possibility to cultivate mid-harvest and late-harvest cultivars, whose fruits can be sold at high prices in China during the Chinese New Year [5].

Southern Chile (37°35′–40°33′ S) has constrained climatic conditions for growing sweet cherry, such as late spring frost, but mainly high precipitation levels at bloom and pre-harvest. Annual rainfall in this location ranges from 1200 to 2800 mm per year [6]. In this regard, rain-induced fruit cracking is one of the main limiting factors for sweet cherry production [7]. Thus, production in this zone is predominantly carried out by using cultivars that are moderately resistant to fruit cracking, such as Regina [7,8,9], and under plastic covers to protect orchards from rains, mainly in the period closest to harvest [7,8].

The use of plastic covers that act as a physical barrier preventing direct water contact on the cherry surface is reported to be an effective strategy to reduce rain-induced cracking in sweet cherry fruits [10]. Nevertheless, several works have shown that plastic covers can also negatively alter fruit quality on parameters such as firmness, fruit resistance to mechanical damage during harvest, handling, and long post-harvest trips [11,12]. Furthermore, color and acidity can be decreased by orchard covering, which is probably explained by the reduced light incidence on covered tree fruits [13]. Interestingly, some studies have shown that fruits produced in the upper zone of the canopy exhibit higher weight, soluble solid contents, and phenolic levels than lower canopy fruits, without significant changes in firmness and acidity [10,14]. These changes could be explained by higher light interception and temperature in the upper zone of the plant [15].

Nowadays, the validation of agronomical strategies capable of overcoming the noxious effects of plastic covers on the quality of sweet cherry fruits cultivated in rainy zones, such as southern Chile, is a key challenge. In this regard, previous studies have shown that adequate potassium (K) nutrition improves several quality traits of fruits, also enhancing shelf-life and shipping quality in fruits such as muskmelon [16,17], cantaloupe [17], and tomato [18]. Adequate K fertilization has demonstrated to increase fruit firmness, caliber and weight, soluble solid content, and acidity in sweet cherries cultivated in Turkey [19]. We recently found that, despite the fact that plastic covers are useful to reduce fruit cracking in Regina sweet cherry fruits cultivated in Southern Chile, covering caused a significant reduction of fruit firmness and acidity at both harvest and post-harvest stages, which were mitigated by using an intensive K foliar regime [20].

The impact of agronomical and technological practices on fresh fruit acidity is relevant, considering that overall consumer acceptance seems to depend primarily on the ratio between sugar and acid concentrations [21]. Changes in the composition of organic acids play a key role in flavor development and can actually affect the chemical and sensory characteristics of fruits, including pH, total acidity, microbial stability, and sweetness [22]. The composition of organic acids can vary in relation to the cultivar, genotype, maturation stages, and weather condition [22,23,24]. Interestingly, it was reported that organic acid composition in fruits can also influence the incidence of fruit cracking, which is mainly due to malic acid in sweet cherries [25]. As far as we know, there are no reports regarding the changes of organic acid composition in sweet cherry fruits cultivated in southern Chile in response to plastic covering and K foliar applications.

Sweet cherries containing high levels of dietary fiber, carotenoids, ascorbic acid, anthocyanins, and phenolic acids are greatly valued by consumers not only due to their taste and color, but also because of their nutritional value and beneficial health effects [26,27]. Sweet cherry phenolic compounds consist mainly of flavonoids, flavan-3-ols, flavonols, and non-flavonoid compounds, such as hydroxycinnamic and hydroxybenzoic acids [28,29,30], which contribute to fruit antioxidant activity [23,29,30,31,32]. Furthermore, sweet cherries are characterized for having anthocyanins as the main phenolics, with 3-rutinoside and 3-glucoside cyanidin derivates as their most abundant compounds [28]. There are a few reports concerning the impact of plastic covers and K supply on fresh fruits phenolics [19], including sweet cherries [33]. This is an interesting research topic, as the increase of synthesis and/or accumulation of antioxidant phenols can positively affect the postharvest life of fresh fruits (reviewed in [34]). Additionally, polyphenolic compounds influence the sensory and organoleptic qualities of fruits such as color, taste, and astringency [35].

This study was aimed towards the influence of plastic covers and pre-harvest K foliar sprays on antioxidant properties as well as the phenolic and organic acid composition in fruit of sweet cherry cv. Regina cultivated in Southern Chile.

## 2. Results

### 2.1. Fruit Technological Quality at Harvest

Results on the impact of covers and K treatments on fruit quality are shown in Table 1. Our results revealed that fruits from covered trees exhibited higher weights compared to cherries harvested from uncovered trees. Furthermore, we found that the weight of the cherry was not significantly affected by the K regime. In addition, data showed that fruits harvested from the upper zone of the canopy and produced on covered trees treated with the conventional K regime showed higher weights than fruits harvested from the lower canopy (Table 1). Moreover, here we found that the size of cherries produced on covered trees was similar to those harvested from uncovered trees. In general, we observed that fruits produced in trees treated with the intensive K regime were bigger compared to those obtained with the conventional K regime; however, this effect was detected only in fruits harvested from the lower canopy zone (Table 1). Similarly, our study revealed that the size of cherries produced in the lower or in the upper zone of the canopy did not significantly differ (Table 2). Interestingly, the use of plastic covers caused a significant decrease in fruit firmness. Furthermore, our observations indicated that fruits treated with additional K doses showed higher firmness in comparison with fruits from trees grown under the conventional K regime. Thus, fruits from uncovered trees subjected to the intensive K regime exhibited ~20% higher firmness than cherries from covered trees grown under the conventional K regime. Furthermore, results showed that fruit firmness was similar among the cherries harvested from both canopy zones (Table 1). Otherwise, the soluble solid content (SSC) of fruits produced under cover was lower than cherries harvested from uncovered fruits; however, this impact was only detected in lower canopy fruits. Fruit SSC increased in response to additional K doses. Therefore, fruits from trees grown under the intensive K regime showed at least ~1.5 Brix more than those from trees grown under the conventional K regime. We also found that SSC was higher in fruits produced in the upper canopy zone compared to fruits harvested from the lower canopy zone (Table 1). Additionally, results indicated that fruit from covered trees exhibited lower fruit titratable acidity (TA) than fruit from uncovered trees. The intensive K regime increased fruit acidity similarly to which occurred with SSC; nonetheless, this intensification was only detected in upper canopy fruits produced in covered trees. In general, TA was similar between fruits from the upper canopy and those produced in the lower zone. Finally, the results indicated that the maturity index (MI) of fruits produced from covered and uncovered trees was similar, being greater in fruits from trees treated with additional K sprays (intensive K regime). This last parameter did not differ between fruits harvested from the upper and lower zone of the canopy (Table 1).

### 2.2. Total Antioxidant Activity and Total Phenolic Content

Both methods used to determine the antioxidant activity of fruits revealed that this parameter was comparable between fruits from covered and uncovered trees. Likewise, results from both procedures indicated that fruit antioxidant activity increased in response to the application of additional K doses; however, these increases were only detected in upper canopy fruits produced by uncovered trees. Therefore, the highest antioxidant activity and total phenolic content were observed in upper canopy fruits produced by uncovered trees grown under the intensive K regime (Table 3). The significance between factors is presented in Table 4.

### 2.3. Phenolic Compound Composition in Fruits

Our study revealed that two anthocyanin molecules were detected in cherry fruits cv. Regina, cyanidin-3-rutinoside and peonidin-3-rutinoside, and cyanidin was the major one. In addition, anthocyanins, flavonols, and hydroxycinnamic acids (HCADs) were also detected in cherry fruits, being anthocyanins, the most abundant compound identified in all samples, followed by HCADs. As for flavonols, a rutinoside derivative of quercetin was detected together with an unidentified compound (*m*/*z* 332 uma). Three HCADs were detected as well, being identified as a quinic acid derivative (*m*/*z* 427.1 uma), product ion 190.9 uma), while two HCADs were tentatively assigned based on their maximum wavelengths, but their molecular and product ions were not detected (Table 5). The concentration of all these molecules was significantly affected by cover and foliar K treatments as well as the position of fruits in the canopy (lower or upper). Therefore, it was found that, when trees were grown under the conventional K regime, fruits from uncovered specimens exhibited higher total anthocyanin concentrations compared to fruits from covered ones. However, this difference was observed only in fruits harvested from the lower canopy, whereas the results from the upper canopy were inverse. Additionally, results indicated that additional K doses increased total anthocyanins in fruits produced in covered trees; nonetheless, this effect was detected only in the lower zone of the canopy. Additional K sprays also increased the anthocyanin concentration in upper canopy fruits, but only in uncovered trees. In fact, the highest anthocyanin concentration was detected in the fruit of the upper canopy in uncovered trees treated with the intensive K regime (Table 5). Regarding flavonols, our study revealed that the concentration of these molecules, quercetin rutinoside and other non-identified flavonols, tended to be higher in fruits produced in uncovered trees. Additionally, the concentration of flavonols increased in response to the application of additional K doses in covered and uncovered trees, which was especially evident for quercetin rutinoside, the major flavonol detected here in cherry fruits. Moreover, in general, flavonol levels were higher in fruits harvested from the upper canopy zone compared to those from the lower canopy zone. Otherwise, hydroxycinnamic acids, which tended to exhibit similar levels in fruits from covered and uncovered tress, increased in response to supplying additional doses of foliar K. Finally, contrasting with anthocyanins and flavonols, phenolic acids tended to be higher in fruits produced in the lower zone of the canopy compared to fruits from the upper canopy. Thus, the high concentration of phenolic acids was detected in upper canopy fruits produced in covered trees treated with the intensive K regime (Table 5). The statistical significance for each factor studied is presented in Table 6.

### 2.4. Organic Acid Composition in Fruits

Results indicated that malic acid was the major organic acid (OA) present in sweet cherry fruits, followed by tartaric, oxalic, and citric acids. The impact of cover and K treatments on OA composition in fruits is shown in Figure 1. When trees were grown under the conventional K treatments, the content of malic acids in fruits from covered and uncovered trees did not vary. However, additional doses of K reduced the content of malic acid in fruits, but only in those produced in uncovered trees. In general, the citric acid content was higher in fruits produced from covered trees compared to fruits from uncovered ones. Otherwise, additional K doses did not affect the content of citric acid. Moreover, results indicated that tartaric, which was detected only in fruits from uncovered trees, decreased in response to the application of additional K doses. When trees were grown under the conventional K regime, the oxalic acid content was significantly higher in cherries from uncovered trees compared to covered ones. However, the application of additional K dosage increased the oxalic acid content in the fruits of covered trees to levels comparable to those detected in the fruits of uncovered trees. Therefore, the lowest content of total OAs was found in fruits produced by uncovered trees treated with the intensive K regime. It is important to highlight that, for all the OAs detected in sweet cherry fruits, the lower concentrations were even detected in fruits harvested from the lower zone of the canopy, except for citric acid (Table 7). Finally, results showed that the proportion of organic acids of fruits from covered trees changed significantly in response to K treatments, being different between fruits harvested in the lower or upper zone of the canopy (Table 7 and Figure 2). Therefore, fruits produced in the upper zone of the canopy exhibited a slightly higher content of malic acid but lower citric acid compared to fruits harvested from the lower zone of the canopy. On the other hand, fruits from trees treated with additional K doses presented a higher content of oxalic acid but lower citric acid content compared to fruits from trees grown under the K^−^ treatment.

### 2.5. PCA Analysis

The PCA provided information about which shoot parameters studied were the most determinant in response to cover and K sprays, giving a partial visualization of the dataset in a reduced dimension. In the present study, the results of PCA (Figure 3) revealed that the first principal component (PC1) accounted for 57.1% of the variance, while principal component 2 (PC2) comprised 29.6% of the variance, indicating a significant differential response of fruit quality and antioxidant properties in response to the different treatments applied. In addition, the loading plots (Figure 3b,c) indicated that PC1 was highly and positively correlated (*r* > 0.8) with total soluble solid content, firmness, and antioxidant activity (using DPPH and ABTS), total flavonols, and oxalic acid concentration, but it was negatively correlated with the concentration of malic and citric acids and total OA. Titratable acidity was the most influential on the second component, which showed a positive correlation (*r* > 0.8) with PC2. PCA identified three groups that varied significantly in terms of fruit quality and antioxidant properties: Group 1, including trees treated with the uncovered/K− and covered/K− treatments; Group 2, comprising trees subjected to covered//K+ treatment; and Group 3, including trees grown under the uncovered/K+ treatments. Group 1 revealed high positive correlations only for malic acid concentration and total OA, while for the other parameters, correlations were negative.

On the other hand, Group 2 exhibited positive correlations with maturity index and citric acid concentrations. Finally, Group 3 showed the highest positive correlations with firmness, solid soluble content, titratable acidity, maturity index, antioxidant activity (using DPPH and ABTS), concentration of all phenolic molecules, and oxalic acid.

## 3. Discussion

The use of plastic covers is one of the strategies used to mitigate the negative impacts of rainy weather on sweet cherry fruit production and quality. Although covers are useful in overcoming the harmful effect of rainfall during the bloom period and before harvest [36], they decrease fruit firmness and acidity [10,20,37], thus reducing fruit quality and overall condition. Otherwise, potassium (K) has been described as an important regulator of several parameters related to fruit yield and quality due to its functions as a catalyst and enzyme activator in different protein reactions and carbohydrate metabolisms [38,39]. In plants, these nutrients play a key role in osmotic regulation, cell division, organic acids neutralization, starch and sugar formation, protein synthesis, and photo-assimilate transport [40]. Recently, we found that an intensive regime of K foliar supply, consisting of 7 sprays during the season, applied from the straw-yellow fruit skin color phenological stage, was useful to increase the quality and condition of sweet cherry fruit produced in covered orchards in the South of Chile [20]. In agreement with our previous work, our current study showed that sweet cherries (cv. Regina) collected from covered trees reduced firmness and acidity but increased caliber and weight compared to uncovered trees. Our observations also revealed that, when trees were treated with the intensive K regime (7 sprays during the season), fruit weight, size, firmness, soluble solid contents, and titratable acidity increased in comparison with the values detected in fruits from trees treated with the conventional K regime (4 sprays during the season), effects that were mainly observed in covered trees.

A complete discussion about the impact of plastic cover and K foliar supply on the quality and condition of sweet cherry was included in Bustamante et al. [20]. Currently, we also investigated the impact of the canopy zone, related to fruit exposition to light, on sweet cherry quality and composition. In this regard, we showed that fruits produced in the upper canopy are more exposed to solar radiation, differing strongly from those harvested in the lower canopy zone. Regarding fruit quality, cherries collected from the upper zone of the canopy exhibited higher weight, soluble solid content, and titratable acidity than fruits from the lower canopy zone, without significant effects on fruit size and firmness. These results coincide with those presented by Kviklys et al. [14], who found that, in apple trees without cover, the fruits present in the upper zone appeared to be higher in weight compared to those harvested in the lower zone. However, Grandi et al. [41] showed that sweet cherry fruits harvested from different canopy zones did not significantly vary, findings that are in agreement with those reported by Blanco et al. [10] for sweet cherry fruits cultivated under covers, collected at different heights of the canopy. Nevertheless, Blanco et al. [10], Kviklys et al. [14], and Grandi et al. [41] did not find differences in the firmness of fruits produced in different zones of the canopy. Regarding the soluble solid content in fruits and in agreement with Blanco et al. [10] for sweet cherries and Kviklys et al. [14] for apples, we found that fruits collected from the upper zone of the canopy exhibited a higher soluble solid content; however, these results were only observed for covered trees, whereas for uncovered ones, the results were contrasting. Finally, our study revealed that the TA of upper canopy fruits and lower canopy fruits did not vary. The differences presented in the fruit quality parameters could be explained by a possible increase in the dry matter content in the fruit of the upper canopy zone, which is related to the total amount of sunlight intercepted, being higher at the top of the canopy [10,15]. In the same way, the increased light exposure and temperature could accelerate the maturation process in the upper area of the canopy in contrast to the lower zone.

In plant species, the relationship between oxidative damage and senescence has been widely recognized [42]. In fact, a decline in fruit postharvest quality is primarily associated with decay and senescence induced by oxidative stress [42]. Therefore, a large pool of antioxidants in fruits can extend their shelf-life and improve postharvest quality by delaying senescence [34,42]. It is noteworthy that sweet cherries contain high levels of fiber, carotenoids, ascorbic acid, and phenolic compounds [26,27]. Phenolic compounds are the main group of phytochemicals in sweet cherries [43], including anthocyanins, flavonols, phenolic acids, and tartaric esters [28,43]. All of these molecules contribute to the total fruit antioxidant activity [23,30,32], playing a protective role against the oxidation processes occurring in fruits, mainly postharvest [44]. However, there are few reports regarding the antioxidant potential and total phenolic content in sweet cherry fruits cultivated in southern Chile and how these parameters can be influenced using plastic covers and nutrient supply.

In this research, the antioxidant capacity of sweet cherry fruits subjected to the different cover and K treatments was measured using two methods: ABTS and DPPH assays. According to Kim et al. [45] and Kelebek and Selli [32], the ABTS assay is applicable to both hydrophilic and lipophilic antioxidant systems, whereas the DPPH assay is useful only to hydrophobic systems. The ABTS assay produced higher values than DPPH assays, making the ABTS assay a better method for expressing the antioxidant capacity of phenolic compounds in fruits [46]. These authors found that the antioxidant capacity of sweet cherries, measured with the ABTS assay, ranged from 3.02 to 7.75 µM TE g^−1^ of FW, depending on the cultivar, whereas the values varied from 2.08 to 4.73 µM TE g^−1^ of FW when the DPPH assay was applied. Accordingly, in the current study, the antioxidant activity values of sweet cherry fruits, regardless of the treatments, were in agreement with previous works [24,32,43,46,47]. In this regard, González-Gómez et al. [47] found that the antioxidant capacity of sweet cherry fruits detected by the ABTS assay ranged from 1.45 to 4.86 mg TE g^−1^ FW. Interestingly, Pissard et al. [48] pointed out that the values of DPPH antiradical efficiency for sour cherry cultivars (reached 5.45 µmol TE g^−1^ FW) were higher than those for sweet cherries (reached 3.69 µmol TE g^−1^ FW). This is in accordance with Martini et al. [49], who found that the antioxidant capabilities of sweet cherry extracts, determined with the ABTS assay, varied from 1.32 to 6.78 μmol ET g^−1^ FW. In addition, here, we found that the antioxidant activity of fruits from covered trees was similar to those harvested from uncovered trees. Furthermore, results indicated that, regardless of the assay, the antioxidant activity of fruits produced in trees subjected to the intensive K regime was higher than those detected in fruits from trees grown under the conventional K regime; however, these increases were only detected in upper canopy fruits produced on uncovered trees.

Furthermore, our results revealed that the TPH levels in the Regina fruits were slightly higher than those detected in previous studies. It was reported that the TPH in sweet cherry fruits varied from 1.0 µg GAE (gallic acid equivalents) g^−1^ FW to 2.4 µg GAE g^−1^ FW, depending on the cultivar, with the higher values detected for cultivar Regina [50,51,52]. Pissard et al. [48] found that the mean of TPH in fruits of sweet cherry cultivars was 1.56 mg GAE g^−1^ FW but ranged from 0.5 to 5.9 mg GAE g^−1^ FW, being lower in sweet cherry cultivars compared to sour cherry cultivars, which was also reported by Ferretti et al. [53] and Kim et al. [54]. Kim et al. [54] reported that TPH levels in sweet and sour cherries ranged from 0.92 to 1.46 and from 1.46 to 3.12 mg GAE g^−1^, respectively, whereas low TPH values in fruits of sweet cherry cultivars reported by Usenik et al. [24] ranged from 0.44 to 0.88 mg GAE g^−1^ FW. Those reported by Ballistreri et al. [55] varied from 0.84 to 1.62 mg GAE g^−1^ FW and those by Hayaloglu and Demir [52] from 0.88 to 2.07 mg GAE g^−1^ FW. Despite that TPH levels in sweet cherry fruits in the present study were higher than the normal range for cherry species, they are significantly lower than those found in blueberries [56], a species that is widely recognized for the high antioxidant capacity and phenolic content. It is worth mentioning that Schmitz-Eiberger and Blanke [57] reported that the TPH of sweet cherries produced in covered trees was higher than those produced on uncovered trees. These authors suggest that the TPH increase shown in sweet cherry fruits grown under covers can hence be explained by the stress of lower light intensity and longer temperature fluctuations under cover, possibly leading to heat and cold stress. The same trend was detected in our study, without significant differences being observed between covered and uncovered trees. Lang et al. [58] found that TPH content was higher in cherries from covered trees, probably due to the combined stress caused by lower light intensity and higher temperature fluctuations. Heat stress under covers, with temperatures rising up to 30 °C on days with intense solar radiation, may induce thermal stress in plants, while night-time temperatures drop to as low as ambient temperature, resulting in larger day–night temperature differences. Heat and cold stress have been shown to induce the synthesis of soluble phenolics by increasing the activity of phenylalanine ammonia lyase (PAL) [59]. Moreover, we found that fruit TPH increased when trees were treated with the intensive K regime compared to those grown under the conventional K supply, which was previously reported by Nguyen et al. [60] for Basil. Similarly, it was reported that AA and TPH increased after K foliar application in fig [61], berries [62], and mango [63]. This effect could be a consequence of an activation of the secondary metabolism increasing the phenylpropanoid pathways in the plant and increasing the ripening process of the fruit [63,64]. In contrast, Yener and Altuntaş [19] showed that K supply reduced TPH and AA in sweet cherry fruits compared to the fruits from control trees.

Furthermore, strong correlations were found between TPH and AA in both red-colored and bicolored cherries [24,65]. Wang et al. [66] found that early ripening cultivars contained higher free phenolic acids, which was positively related to the remarkable antioxidant properties of sweet cherry fruits. The same authors stated that DPPH testing showed a strong positive correlation with TPH and specific phenolic compounds such as free epicatechin and conjugated 2,3,4-trihydroxybenzoic acids. As expected, we found that TPH was also higher in the fruits harvested from the upper canopy zone compared to the lower zone. These results coincide with those previously reported by Kviklys et al. [14] for apples, who found that fruits collected from the upper canopy zone exhibited a higher concentration of anthocyanins and TPH. Thus, our results showed that the higher AA was detected in fruits harvested from the upper zone of the canopy of uncovered trees treated with the intensive K regime.

Phenolic molecules in sweet cherry fruits include flavan-3-ols, flavonols and non-flavonoids such as hydroxycinnamic and hydroxybenzoic acids, and anthocyanins [28,29,30,55,67]. According to Martini et al. [49], most of the studies carried out so far identified and quantified sweet cherry phenolics by using high performance liquid chromatography coupled with a photodiode array detector (HPLC-DAD) [28,55], but very few studies applied mass spectrometry for the qualitative analysis and characterization [67,68], as it was used in the current study. The amount of individual phenolic compounds in sweet cherries strongly depends on cultivars [67] and on ecological factors, rootstock, and cultural practices such as irrigation, pruning, and fertilization [31,69,70]. As far as we know, there are very few reports about both the phenolic composition of sweet cherry fruits produced in southern Chile and how this can be influenced by covers and K supply.

Currently, we detected three phenolic acids in sweet cherry fruits: two non-identified hydroxycinnamic acids and a quinic acid derivative. Interestingly, this was the major type of phenolic compound detected in Regina fruits. In this regard, Martini et al. [49] showed that hydroxycinnamic acids were quantitatively the most important class of phenolic compounds in almost all sweet cherry cultivars. Caffeoylquinic and coumaroylquinic acids have been described as the major hydroxycinnamic acids in sweet cherry fruits [28,55]. Jakobek et al. [71] found that, in spite of the rootstock, the major phenolic acids detected in fruits of sweet cherry were neochlorogenic acid (18–50 mg 100 g^−1^ FW), chlorogenic acid (19–62 mg 100 g^−1^ FW), and *p*-coumaric acid derivatives (15–125 mg 100^−1^ FW). These findings are in accordance with other studies [24,54,72,73]. Nonetheless, Mozetič et al. [70] found that the chlorogenic acid content in fruits of cultivars Hartland, Hedelfinger, Black Gold, and Regina was in the range of 1.3–6.5 mg 100 g^−1^ FW, being so much higher (ranged from 75–507 mg 100 g^−1^ FW) in fruits of cultivars Lambert, Bing, Stella Compact, Napoleon, and Petrovka. Neochlorogenic and chlorogenic acids were also reported by Liu et al. [65] as the main hydroxycinnamic acids in sweet cherries. According to Kim et al. [54], the *p*-coumaric acid derivatives in fruits of sweet cherry are in the range of 10–68 mg kg^−1^ FW, depending on the cultivar. In the same way, Hayaloglu and Demir [52] showed that concentrations of phenolic compounds in sweet cherries are significantly influenced by cultivars, the main being neochlorogenic and *p*-coumaroylquinic acids, which were detected at concentrations ranging from 24.0 to 146.6 mg 100 g of FW and 17.3 to 98.7 mg 100 g^−1^ FW, respectively. According to the same authors, chlorogenic acid was also detected in sweet cherries but at significantly lower concentrations (until 5.09 mg 100 g^−1^ FW) than neochlorogenic and *p*-coumaroylquinic acids. Additionally, gallic acid and 2,5-dihydroxybenzoic acid, also detected in sweet cherry fruits, varied between 0.68 and 10.64 mg 100 g^−1^ FW and from 0.46 to 1.64 mg 100 g of FW, respectively, depending on the cultivar [52]. In the same regard, Wang et al. [66] showed that, in the free phenolic (FP) fractions of sweet cherry fruits, a total of 8 phenolic compounds were found, with neochlorogenic acid being the most abundant acid (from 28.6 to 69.5% of the FP), while chlorogenic acid was the least abundant acid in all samples (ranging from 4.4 mg100 g^−1^ DW to 20.0 mg 100 g^−1^ DW). According to the same authors, (i) the FP fraction also contained small amounts of *p*-hydroxybenzoic and vanillic acid; (ii) quinic and chlorogenic acids were the major conjugated compounds in the early ripening cultivars, whereas in the late ripening ones, caffeic acid was the most abundant, followed by 2,3,4-trihydroxybenzoic, *p*-coumaric, and pyrocatechuic acids; (iii) the subtotals of conjugated phenolics in the late ripening cultivars were higher than those in the early ripening cultivars, mainly due to the higher caffeic acid. Our results indicated higher hydroxycinnamic acid concentrations than the values reported by Usenik et al. [24], Gonçalves et al. [29], Kelebek and Selli [32], and Serradilla et al. [74]. The reason for the difference is that phenolic levels, in addition to the cultivar genotype, depend on the ripening stage, environmental conditions, and extraction process [29,34]. Furthermore, our study revealed that the impact of covers and K supply on the amount of hydroxycinnamic acids in fruits was unclear; nonetheless, these tended to be higher in fruits from covered trees, and there was no clear response to additional doses of foliar K. Moreover, phenolic acids tended to be higher in fruits produced in the lower zone of the canopy compared to fruits from the upper canopy. Thus, the higher hydroxycinnamic acid level was detected in upper canopy fruits produced in covered trees treated with the intensive K regime. As far as we are concerned, there are no reports about the influence of covers and K supply on the phenolic acid content and composition of sweet cherry fruits.

In this study, we found quercetin rutinoside and a non-identified flavonol as the major flavonols in fruits of cv. Regina, which ranged from 7.3 to 10 mg 100 g^−1^ FW and from 2.7 to 4.9 mg 100 g^−1^ FW, depending on the treatment. It was reported that epicatechin and quercetin-3-rutinoside are the main flavonoids in sweet cherries [68]. Jakobek et al. [71] found that quercetin-3-rutinoside in sweet cherries varied from 8 to 37 mg kg^−1^ FW, whereas Kim et al. [54] also detected this compound in fruits of some sweet cherry cultivars, including Regina, but at slightly higher concentrations (11.4 to 46.9 mg kg^−1^ FW). Contrastingly, Wang et al. [66] pointed out that epicatechin was found to be the main compound detected in sweet cherries with smaller amounts of catechin, which is consistent with previous findings [29,32]. These authors found that levels of epicatechin ranged from 6.3 to 86.6 mg 100 g^−1^ DW, which is in accordance with González-Gómez et al. [47] and Liu et al. [65] but lower than those reported by Kelebek and Selli [32] and Hayaloglu and Demir [52]. According to Hayaloglu and Demir [52], (-)-epichatechin and rutin were among the main flavanols found in sweet cherry fruits. The concentration of (-)-epicatechin from 5.27 to 14.67 mg 100 g of FW was determined by the cultivar. Serradilla et al. [74] and Liu et al. [66] detected rutin at levels from 1.34 to 3.13 mg 100 g^−1^ FW in sweet cherry fruits. Interestingly, the study of Wang et al. [66] revealed that the higher levels of epicatechin and catechin were detected in early maturity cultivars.

Anthocyanins have been reported to be the main phenolic in fruits from some sweet cherry cultivars, where the major compounds are aglycon cyanidin bound to the glycosides 3-rutinoside and 3-glucoside, and the minor ones are 3-rutinoside of peonidin and pelargonidin [28,55,75]. In fact, Ballistreri et al. [55] found that cyanidin-3-O-rutinoside accounted for 70% to 90% of the total anthocyanin content in sweet cherries, which ranged from 5.69 to 68.28 mg CGE (cyanidin 3-glucoside equivalents) 100 g^−1^ FW, followed by cyanidin-3-O-glucoside (from 10% to 24%). Likewise, Hayaloglu and Demir [52] reported that, among the six anthocyanins detected in cherries, cyanidin-3-O-rutinoside was the principal molecule. The same anthocyanins were found to be predominant in sweet cherries by other authors [24,29,65,74,76]. Gao and Mazza [28] and Kelebek and Selli [32], reported that the total content of anthocyanins in sweet cherries ranged from 47 to 128.51 mg 100 g^- 1^ FW for dark cultivars and from 2 to 41 mg 100 g^−1^ FW for less-colored cultivars. These results are in accordance with those obtained in the present study, which revealed that the total anthocyanin concentration ranged between 43.7 and 213.7 mg 100 g^−1^ FW, as Regina is a dark cultivar. Regarding the anthocyanin composition, we found that cyanidin 3-rutinoside and peonidin 3-rutinoside were detected in Regina fruits, cyanidin being the major one, which accounted for at least 80% of the total anthocyanin content, regardless of the treatments. Similar amounts of anthocyanins were reported by Gonçalves et al. [29], Kim et al. [54], and Serradilla et al. [74]. Furthermore, our observations showed that the anthocyanins were significantly higher in fruits of the upper zone of the canopy compared to fruits of the lower canopy. Additionally, we showed that the concentration of anthocyanins was similar in fruits produced in covered and uncovered trees. This is consistent with the results obtained by Usenik et al. [24], who did not find any influence of covers on the phenolic composition of sweet cherry fruits. Additionally, our results indicated that total anthocyanin concentrations in fruits significantly varied by the impact of the K regime applied. In this way, we also found that fruits from trees treated with the intensive K regime exhibited higher total anthocyanin concentrations compared to fruits from trees grown under the conventional K regime. Other authors reported that the increase in K supply raised the concentration of anthocyanins in pomegranates, apples, and wine grapes [33,77,78]. On the other hand, Solhjoo et al. [33] displayed a positive correlation between the K and anthocyanin concentrations in apple fruits. According to Delgado et al. [77], the change in total anthocyanin concentration in table grapes in response to the K supply was due to the effect between the balance of N and K in the tissues, describing that low N in conjunction to high K in fruit tissues reduces the formation of anthocyanins; on the contrary, high or moderate N concentrations in conjunction with high K would generate an increase in anthocyanins in fruits. It has been reported that K is a key element in the anthocyanin pathway, acting as a cofactor in the activation of some specific enzymes such as UDPGalactose flavanoide-3-O-glicosiltransferase [79].

Organic acids are the second main family of organic compounds found in sweet cherries, after carbohydrates. They are important components of cherries in terms of their impact on the taste of the fruit [24,80]. Our results on the composition of organic acids in Regina fruits revealed that the sum of organic acids detected in Regina fruits (malic, citric, tartaric, and oxalic) ranged from 1.19 to 1.51 g kg^−1^ of FW, depending on the treatment. These values were around 10-times lower than those reported by Kelebek and Selli [32], which could be explained by the reduced acidity of sweet cherry fruits produced in southern Chile previously reported by Bustamante et al. [20] and Villavicencio et al. [81]. In addition, we found that malic acid was the most abundant organic acid in fruits, followed by tartaric acid, oxalic acid, and citric acids. In this regard, Kelebek and Selli [32] showed that malic acid was the main organic acid in sweet cherry fruits (8.54–10.02 g kg^−1^ of FW), whereas the sum of organic acids ranged from 12.01 to 14.17 g kg^−1^ of FW, depending on the cultivar. These results are in agreement with those reported by Girard and Kopp [82], who also pointed out that malic acid varied widely (5.02–9.48 g kg^−1^ of FW) among cultivars. According to the same authors, citric acid was the second most abundant organic acid in the analyzed sweet cherry cultivars, ranging from 1.39 to 2.69 g kg^−1^ of FW, whereas the amounts of shikimic acid and fumaric acid ranged from 0.51 to 2.05 g kg^−1^ of FW and 0.36 to 1.10 g kg^−1^ of FW, respectively. Furthermore, we found that fruits produced in covered trees exhibited higher citric acid concentrations. These results agree with Usenik et al. [24], who reported that covered sweet cherry plants cv. Regina showed a tendency to increase citric acid but to maintain the concentration of malic acid in fruits at harvest compared to fruits from uncovered trees. Our results also revealed an unclear effect of K regimens on malic acid concentration in sweet cherry trees. Contrarily, Nagy et al. [83] found that the two-time application of K during the season (25 and 35 days after full bloom) generated a higher concentration of citric and malic acids compared to control treatments in “Germersdorfi 3” cherry plants (without K supply). The above shows that the behavior of organic acids in cherry trees can vary depending on the cultivar and the light conditions presented in the orchards. According to Marschner [40], the accumulation of organic acid anions (such as malic acid) in plant tissues is often the consequence of K transport without accompanying anions, because, in charge compensation, K is the dominant cation for counterbalancing immobile anions in the cytoplasm, chloroplasts, and quite often also for mobile anions in vacuoles, the xylem, and the phloem.

Regarding the effect of the treatments on the different canopy sections, our results revealed a higher concentration of total organic acids in the upper canopy zone compared to the lower canopy. We similarly found that additional doses of K increased total organic acid, but only in the lower canopy section. This observation can be explained by the higher TA (titratable acidity) detected in fruits from trees subjected to the intensive K regime discussed above.

## 4. Materials and Methods

### 4.1. Plant Material, Treatments, and Experimental Design

A field study was carried out during the season 2018/2019, in a sweet cherry (*Prunus avium* L.) cultivar Regina commercial orchard, grafted on Gisela 6 rootstock, located in the La Araucanía Region (Perquenco locality; 38°24′9.39″ S, 72°30′49.38″ W; 250 m.a.s.l.). In this orchard, cherry trees were planted in 2012, using a 4.5 m × 1.6 m planting design (east-west row orientation), reaching a planting density of 1389 trees ha^−1^. Regarding the climate conditions of this location, the minimum, maximum, and average temperatures from veraison to harvest period (November to February) were 6.8 °C, 22 °C, and 14.5 °C, respectively, whereas the average relative humidity (RH) was 69%, and the accumulated precipitation was 148 mm during the same period (data recorded by the San Sebastián Station, INIA). It is noteworthy that this location has a temperate oceanic rainy climate with an average annual precipitation of 1200 mm [6].

In this orchard, the plastic covers used consisted of a movable high-density polyethylene (HDPE) plastic in gable form that had the following properties: 4 m width, an average density of 160 g m^−2^, 88–90% of total light transmission, and 60–65% diffuse light transmission. The cover was deployed (closed) from the anthesis to the phenological stages of fruit set and then 2 weeks before harvest.

### 4.2. Treatments

Covered and uncovered trees were treated with potassium (K) foliar sprays by using a bag bomb (Cifarelli^®^, Model L3EDA, Voghera, Italy). In each K spray, 4 L ha^−1^ of the chelated fertilizer Nutripotasio Plus (ADAMA^®^; 38% of K_2_O; di-luted in water) was applied to reach a final concentration of 300 mL hL^−1^. For the treatment identified as the conventional K regime, foliar K sprays were weekly applied from the straw-yellow fruit skin color phenological stage, for 4 weeks, reaching a total of 4 sprays during the season, whereas for the treatment identified as the intensive K regime, 3 additional sprays were applied weekly, from the last application of the conventional K regime to 1 week prior to harvest, reaching a total of 7 sprays. In addition, to evaluate the impact of the fruit light exposure on the different parameters studied, fruits were separately harvested from the upper canopy (height > 1.2 m) and the lower canopy (height < 1.2 m). Then, in this study, a total of 8 treatments were assessed, considering the interaction of covering × K regime × canopy zone.

### 4.3. Experimental Design

The experimental design used was a split-plot. One plot consisted of covered trees, while the other consisted of uncovered trees. Each plot was divided into five blocks per K treatment. Each block corresponded to a row of 4 consecutive trees. Fruit analyses were carried out only on fruits collected from the 2 central trees of each block, whereas the 2 remaining trees were considered border trees.

### 4.4. Evaluation of Fruit Quality at Harvest

A total of 200 fruits per tree were collected, considering both tree exposures and 100 cherries per each canopy zone. Once harvested, fruits were stored in 1 kg plastic bags and transported (4 °C) to the Fruit Crops Physiology and Quality Laboratory of La Frontera University. Then, fresh cherries were stored at 4 °C prior to measuring quality and condition parameters within 48 h after harvest. Thus, 5 independent samples were analyzed per treatment. The physical quality parameters of weight, equatorial diameter (size), and firmness of 35 fresh fruits per sample were measured, using an analytical balance and a texture-meter (FirmPro, Happyvolt, Santiago, CL, USA). This equipment automatically measures fruit caliber and firmness at the same time, which has been validated for fruit firmness determination in sweet cherry cultivar Lapins [84]. Furthermore, total soluble solids (TSS), titratable acidity (TA), and maturity index (MI) were determined. Briefly, TSS was analyzed in the fruit juice using a thermo-compensated digital refractometer (ATAGO, Mod. PAL-BX I ACID F5, Saitama, Japan) and expressed as °Brix. In addition, TA was determined by the volumetric titration method with sodium hydroxide (0.1 N) using an automatic titrator (HANNA Mod. HI-84532, Woonsocket, RI, USA) and was expressed as percentage (%) of malic acid. Finally, the maturity index was calculated as the ratio between TSS and TA. A trained panel qualitatively and visually estimated the most frequent fruit condition defects, including cracking, browning pedicels, and pitting, which were previously analyzed to determine physical parameters.

### 4.5. Antioxidant Activity Determinations in Fruits

An amount of 245 μL of ABTS 7.5 mM and 5 μL of fruit extract or TROLOX standard was added for the Trolox Equivalent Antioxidant Capacity (TEAC) method. The samples were incubated at 30 °C for 30 min. Subsequently, measurements were carried out at 734 nm, and the results were expressed as TROLOX equivalents [85]. In addition, the measurement of antioxidant activity by the DPPH method was based on that reported by Maldonado et al. [86], with minor modifications. Briefly, 240 μL of 0.1 mM DPPH radical was dissolved in methanol, and the first absorbance was measured. Then, 10 μL of the sample or standard was added and incubated for 30 min in the dark, and then the second absorbance was measured. Measurements were carried out at 517 nm, and results were expressed as TROLOX equivalents. Epoch microplate reader (Bio Tek, Winooski, VT, USA) was used for absorbance lectures.

### 4.6. Analysis of Phenolic Composition in Fruits

The biochemical analyses were carried out based on 12 biological repetitions (fruits) per treatment. Two grams of fruits were crushed, and 4 mL of the extraction solvent ethanol: acetic acid 97: 3 *v*/*v* was added. Subsequently, the samples were sonicated with an ultrasonic processor at 130 Watt (Sonics & Materials, Newtown, CT, USA) for 60 s at 40% amplitude and then shaken for 30 min at 200 g. Finally, they were centrifuged (Lab companion, Seoul, Korea) for 10 min at 4000 g. Consequently, the supernatant was stored at −20 °C until the analysis. The identification of phenolic compounds was carried out according to the method described by Parada et al. [85] using high-performance liquid chromatography (HPLC) with diode array detection (DAD) equipped with an LC-20AT quaternary pump, a DGU-20A5R degassed unit, a CTO-20A oven, a SIL-20A autosampler, and a UV visible diode array detector (SPD M20A) (Shimadzu, Tokyo, Japan). Control and data collection were carried out using Lab Solutions software (Shimadzu, Duisburg, Germany). Identity assignments were performed by using an HPLC-DAD system coupled to a QTrap 3200 LC/MS/MS mass spectrometer from Applied Biosystems MDS Sciex system (Foster City, CA, USA). Data were collected using Analyst software (v. 1.5.2) (SCIEX, Woodlands Central Indus. Estate, Singapore) for MS/MS analysis. The chromatographic separation method for the determination of phenolic compounds was carried out based on the method reported by Parada et al. [85] with modifications, using a Kromasil Classic-Shell C_18_ column (100 × 4.6 mm, 2.5 μm) and a Novapak Waters C_18_ pre-column (22 mm × 3.9 mm, 4 μm). The extracts were injected at a 40 °C with a flow rate of 0.55 mL min^−1^. The mobile phases used were 92:3:5 (v:v:v) of water, acetonitrile, and formic acid (mobile phase A) and 45:50:5 (v:v:v) of water, acetonitrile, and formic acid (mobile phase B), with an elution gradient from 6 to 50% B in 30 min. Detection wavelengths were 520 nm for anthocyanins, 320 nm for hydroxycinnamic acids, and 360 nm for flavonols, using cyanidin-3-glucoside, chlorogenic acid, and quercetin, respectively, as standards for external calibration.

### 4.7. Measuring Total Phenolic Compounds in Fruits

The following reagents were added in the following order to a microtube: 15 μL of standard or extract, 750 μL of deionized water, 75 μL of Folin-Ciocalteu reagent, 300 μL of sodium carbonate 20% *m*/*v*, and 360 μL of deionized water. The solutions were incubated at 20 °C for 30 min in the dark; then, 250 μL of the solution was added to a 96-well plate, and the absorbance was obtained at 750 nm [85]. The determinations were carried out using gallic acid as the standard. Epoch microplate reader (Bio Tek, Winooski, VT, USA) was used for absorbance lectures.

### 4.8. Analysis of Low Molecular Weight Organic Acid Composition in Fruits

Fresh samples of sweet cherry (10 g) were homogenized with 20 mL of distilled water. The fruit puree was centrifuged (Labnet, Prism, NJ, USA) at 3000 g for 10 min. The supernatant was filtered through 0.25 μm Millipore filters and frozen by freeze-drying [22]. The residue was re-dissolved in 500 μL of deionized sterile water (HPLC grade) for HPLC injection and then filtered (0.22 μm). The chromatographic separation was performed in an HPLC system (Jasco LC-Net II/ADC) equipped with a photodiode array detector (DAD) (Jasco MD 2015 Plus) according to Millaleo et al. [87]. The flow rate was 1 mL min^−1^, and injected samples were detected at a wavelength of 210 nm. Organic acid anions (oxalic, malic, citric, and tartaric acids) were identified by comparing the retention times and adding standards for each organic acid anion.

### 4.9. Statistical Data Analysis

The data were analyzed using linear mixed models (Mlmix) with a factorial model (potassium ×covering) at a significance level of 95% (*p*-value < 0.05). If there were statistically significant differences, the means were compared using a Fisher’s LSD multiple comparison test at 95% (*p*-value < 0.05). Different lowercase letters per row indicate statistically significant differences between treatments with interaction between the three factors. Asterisks indicate the significance of the factors for each variable (* *p* < 0.05, ** *p* < 0.01, *** *p* < 0.005). Additionally, a principal component analysis (PCA) was conducted to identify the variables that explained the relationship between fruit quality parameters and the applied treatments. The proportion variance explained (estimated as the sum of squares of the treatment over the total sum of squares) for the factors involved in this study was analyzed. The statistical software INFOSTAT (version 2017, Argentina) was used for all data analyses.

## 5. Conclusions

In accordance with our previous studies, we found that sweet cherries (cv. Regina) produced in covered trees showed reduced firmness and acidity but increased caliber and weight compared to uncovered trees. It is worth mentioning that the covered trees treated with the intensive K regime (7 sprays) presented an increase in all quality fruit parameters compared to fruits from trees with 4 sprays. Similarly, we observed an effect of light exposure on the variable the upper canopy zone, which presented a better condition. Regarding sweet cherry composition, we detected three phenolic acids in sweet cherry fruits: two non-identified hydroxycinnamic acids and a quinic acid derivative, the latter being the major type of phenolic compound detected in Regina fruits. Quercetin rutinoside and a nonidentified flavonol were detected as the major flavonol molecules in fruits, while cyanidin 3-rutinoside and peonidin 3-rutinoside were the major anthocyanins, with cyanidin accounting for at least for 80% of the total content. In the same way, the potassium effect was evidenced over each evaluated parameter, showing a positive tendency between the potassium sprays and the concentration of phenolic compounds. Likewise, the organic acid concentration was increased by the intensive K regime in covered trees, which could be explained by a lower degradation of these acids in the maturation phase of the fruit; thus, it could be the reason for the increased titratable acidity of this treatment. In summary, in this study, the potassium application from pit hardening to harvest was evidenced as an effective tool used to improve the quality, phenolic composition, and organic acids of fruit harvested from sweet cherry trees under plastic covers. Likewise, potassium sprays were an effective management technique to possibly improve the organoleptic quality (by increasing organic acids and titratable acidity) in the sweet cherry fruit grown in the south of Chile.

## Figures and Tables

**Figure 1 plants-12-00050-f001:**
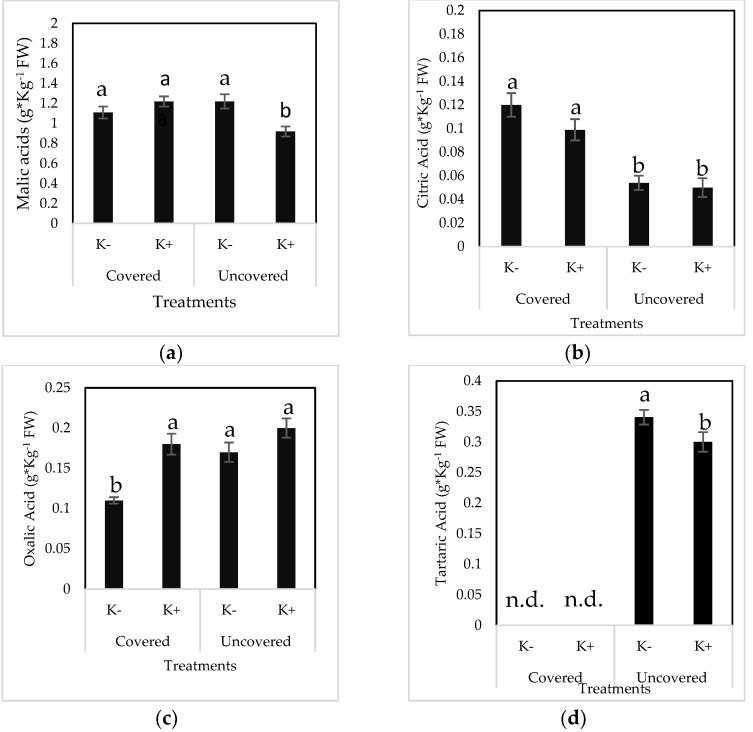
The effect of two potassium spray treatments, conventional K regime (4 sprays during the season: K- treatment) or intensive K regime (7 sprays during the season: K+ treatment), on malic acid (**a**), citric acid (**b**), oxalic acid (**c**), tartaric acid (**d**), and total acid concentration (**e**) in sweet cherry fruits (cv. Regina) from covered and uncovered trees. Statistically significant differences between treatments are represented by different letters above bars, based on the LSD Fisher multiple range test (*p* ≤ 0.05). Error bars indicate the standard error for each dataset. Abbreviation: OA (organic acids).

**Figure 2 plants-12-00050-f002:**
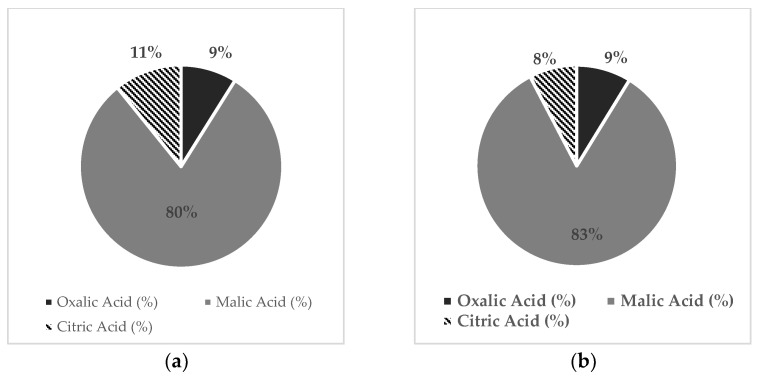
Proportion of organic acid in sweet cherry (cv. Regina) fruits subjected to two foliar potassium (K) treatments: conventional K regime (4 sprays during the season: K− treatment) or intensive K regime (7 sprays during the season: K+ treatment). Fruits harvested from the different zones of the canopy were analyzed: (**a**) lower canopy fruits from trees treated with the conventional K regime; upper canopy fruits from trees treated with the conventional K regime (**b**); lower canopy fruits from trees treated with the intensive K regime (**c**); upper canopy fruits from trees treated with the intensive K regime (**d**).

**Figure 3 plants-12-00050-f003:**
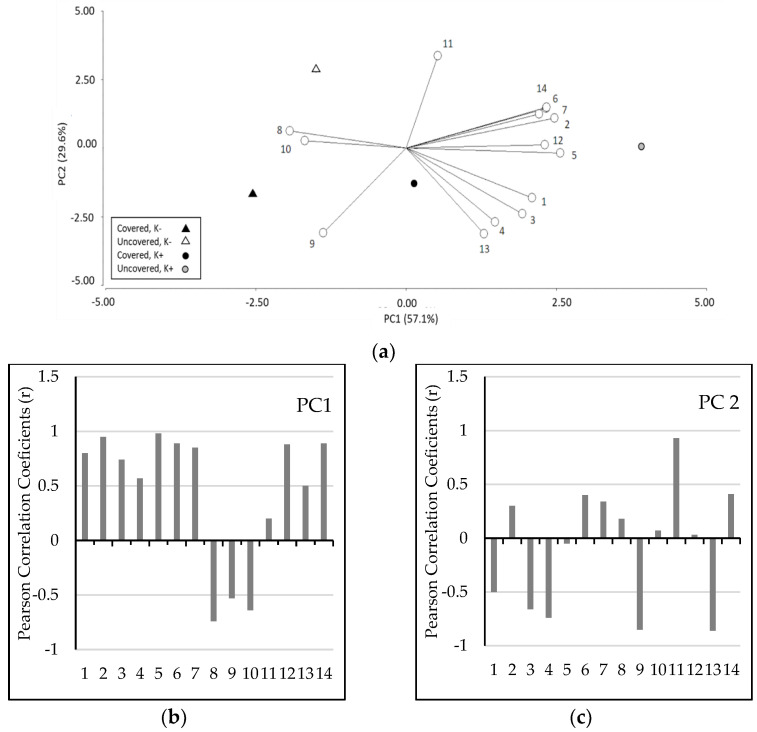
Principal component analysis (PCA): (**a**) scatter plot of principal component 1 against principal component 2 and loading plots of PC1 (**b**) and PC2 (**c**), representing the quality parameter, antioxidant activity, phenolic content, and concentration of phenolic and organic acids in fruits of sweet cherry cultivated in a commercial orchard of southern Chile, subjected to different cover and k treatments. Fruits from both covered and uncovered trees subjected to two foliar conventional K regime (4 sprays during the season: K- treatment) or intensive K regime (7 sprays during the season: K+ treatment) were evaluated. The white points in the scatter plot are total anthocyanins (1), total flavonols (2), total hydroxycinnamic acids (3), total phenols (4), antioxidant activity (using DDPH) (5), antioxidant activity (using ABTS) (6), oxalic acid (7), malic acid (8), citric acid (9), total OA (10), titratable acidity (11), total solid soluble content (12), maturity index (13), and firmness (14). The results of tartaric acid were excluded due to the fact that this OA was detected only in fruits from uncovered trees.

**Table 1 plants-12-00050-t001:** Effects, at harvest, of two K spray treatments on the quality of sweet cherry (cv. Regina) fruits produced from covered and uncovered trees: conventional K regime (4 sprays during the season: K− treatment) or intensive K regime (7 sprays during the season: K+ treatment). Fruits from the upper and lower zone of the canopy were separately analyzed. The statistical significance between canopy zones for each treatment is also shown. Abbreviations: SSC (soluble solid content); TA (titratable acidity).

Quality Parameter	Canopy Zone	Treatments			
		Uncovered Trees		Covered Trees	
		K−	K+	K−	K+
Fruit weight (g)	Lower	9.5 ± 0.2 Ba	9.5 ± 1.0 Ba	9.9 ± 0.2 Ab	10.3 ± 0.2 Aa
	Upper	9.9 ± 0.2 Ba	9.8 ± 0.2 Ba	10.4 ± 0.2 Aa	10.3 ± 0.2 Aa
	Significance	n.s.	n.s.	*	n.s.
Fruit size (mm)	Lower	26.6 ± 0.3 Ba	27.3 ± 0.2 Aa	27.1 ± 0.3 Ba	27.7 ± 0.3 Aa
	Upper	26.8 ± 0.3 Aa	27.3 ± 0.2 Aa	27.1 ± 0.2 Aa	27.3 ± 0.3 Aa
	Significance	n.s.	n.s.	n.s.	n.s.
Firmness (g mm^−1^)	Lower	341.3 ± 10.1 Ba	373.3 ± 6.9 Aa	293.6 ± 6.8 Ca	315.2 ± 7.1 Ca
	Upper	323.4 ± 10.3 Ba	369.2 ± 8.3 Aa	296.7 ± 7.8 Ca	311.6 ± 9.7 BCa
	Significance	n.s.	n.s.	n.s.	n.s.
SSC (Brix)	Lower	17.5 ± 0.9 Ba	19.1 ± 0.5 Ab	15.9 ± 0.7 Cb	18.1 ± 0.5 Bb
	Upper	18.5 ± 0.9 Bb	19.9 ± 0.3 Aa	17.9 ± 0.5 Ba	20.2 ± 0.5 Aa
	Significance	**	*	***	***
TA (% of malic acid)	Lower	1.1 ± 0.0 Aa	1.0 ± 0.0 Aa	0.9 ± 0.0 Ba	0.9 ± 0.0 Bb
	Upper	1.1 ± 0.0 Aa	0.9 ± 0.0 BCa	0.9 ± 0.0 Ca	1.0 ± 0.0 ABa
	Significance	n.s.	n.s.	n.s.	*
Maturity index	Lower	16.7 ± 0.9 Ba	18.8 ± 0.6 Aa	18.4 ± 1.0 Ba	20.5 ± 0.9 Aa
	Upper	16.9 ± 0.6 Ba	21.0 ± 1.0 Aa	19.9 ± 0.6 Aa	19.4 ± 0.3 Aa
	Significance	n.s.	n.s.	n.s.	n.s.

Means ± standard errors (SE) are shown. Statistically significant differences between treatments for each canopy zone are represented by different uppercase letters (horizontally), whereas differences between canopy zone for each K treatment are represented by vertical lowercase letters, based on the LSD Fisher multiple range test (*p* ≤ 0.05). Asterisks indicate a significant interaction between the canopy zone in each treatment (* *p* ≤ 0.05. ** *p* ≤ 0.01. *** *p* ≤ 0.005). n.s. indicates non-statistical significance.

**Table 2 plants-12-00050-t002:** The statistical significance of each factor and their interaction with the analysis of fruit quality.. Abbreviations: SSC (soluble solid content); TA (titratable acidity).

Quality Parameters	Canopy Zone	Potassium	Cover	Potassium × Cover
Fruit weight	Lower	n.s.	**	n.s.
	Upper	n.s.	**	n.s.
Fruit size	Lower	**	n.s.	n.s.
	Upper	n.s.	n.s.	n.s.
Firmness	Lower	**	***	n.s.
	Upper	***	***	n.s.
SCC	Lower	***	***	n.s.
	Upper	***	n.s.	n.s.
TA	Lower	n.s.	***	n.s.
	Upper	n.s.	n.s.	***
Maturity index	Lower	*	n.s.	n.s.
	Upper	*	n.s.	**

Asterisks indicate a significant interaction between the factors (* *p* ≤ 0.05. ** *p* ≤ 0.01. *** *p* ≤ 0.005). n.s. indicates non-statistical significance.

**Table 3 plants-12-00050-t003:** Effects of two potassium spray treatments, conventional K regime (4 sprays during the season: K− treatment) or intensive K regime (7 sprays during the season: K+ treatment), on antioxidant activity and total phenolic content in whole fruits of sweet cherry (cv. Regina) produced in covered and uncovered trees. Fruits from the upper and lower zone of the canopy were separately analyzed. The statistical significance between canopy zones for each treatment is also shown. Abbreviations: TE (Trolox equivalents); GAE (gallic acid equivalents); ABTS (2,2- Azino bis (3- ethylbenzothiazolin-6- sulfathan); DPPH (1,1-diphenyl-2-picrylidracil); FW (fresh weight).

Compounds	Canopy Zone	Treatments			
		Uncovered Trees		Covered Trees	
		K−	K+	K−	K+
Antioxidant activity (using DPPH) (µmol TE g^−1^ FW)	Lower	1.6 ± 0.0 Aa	1.5 ± 0.0 Ab	1.5 ± 0.1 Aa	1.6 ± 0.1 Aa
	Upper	1.5 ± 0.1 Ba	1.8 ± 0.0 Aa	1.5 ± 0.1 Ba	1.6 ± 0.1 Aa
	Significance	n.s.	***	n.s.	n.s.
Antioxidant activity (using ABTS) (µmol TE g^−1^ FW)	Lower	2.4 ± 0.1 Aa	2.3 ± 0.1 Aa	2.3 ± 0.1 Aa	2.4 ± 0.1 Aa
	Upper	2.4 ± 0.1 Ba	3.0 ± 0.1 Aa	2.3 ± 0.1 Ba	2.3 ± 0.1 Ba
	Significance	n.s.	n.s.	n.s.	n.s.
Total phenolic content (µg GAE g^−1^ FW)	Lower	2.5 ± 0.1 Ba	2.9 ± 0.2 Aa	2.8 ± 0.4 Ba	3.4 ± 0.4 Aa
	Upper	2.5 ± 0.3 Aa	3.3 ± 0.4 Aa	2.8 ± 0.2 Aa	2.9 ± 0.2 Aa
	Significance	n.s.	n.s.	n.s.	n.s.

Means ± standard errors (SE) are shown. Statistically significant differences between treatments for each canopy zone are represented by different uppercase letters (horizontally), whereas differences between canopy zones for each K treatment are represented by vertical lowercase letters, based on the LSD Fisher multiple range test (*p* ≤ 0.05). Asterisks indicate a significant interaction between the canopy zone in each treatment (*** *p* ≤ 0.005). n.s. indicates non-statistical significance.

**Table 4 plants-12-00050-t004:** The statistical significance of each factor and their interaction for the analysis of antioxidant activity and total phenolic content. Abbreviations: TE (Trolox equivalents); GAE (gallic acid equivalents); ABTS (2,2- Azino bis (3- ethylbenzothiazolin-6- sulfathan); DPPH (1,1-diphenyl-2-picrylidracil); FW (fresh weight).

	Zone	Significance		
Compounds		Potassium	Cover	Potassium × Cover

Antioxidant activity (using DPPH)	Lower	n.s.	n.s.	n.s.
	Upper	*	n.s.	n.s.
Antioxidant activity (using ABTS)	Lower	n.s.	n.s.	n.s.
	Upper	**	***	*
Total phenolic content	Lower	*	*	n.s.
	Upper	n.s.	n.s.	n.s.

Asterisks indicate a significant interaction between the factors (* *p* ≤ 0.05. ** *p* ≤ 0.01. *** *p* ≤ 0.005). n.s. indicates non statistical significance.

**Table 5 plants-12-00050-t005:** Effect of two potassium spray treatments on phenolic compound produced in covered and uncovered trees: conventional K regime (4 sprays during the season: K− treatment) or intensive K regime (7 sprays during the season: K+ treatment). Fruits from the upper and lower zone of the canopy were separately analyzed.

Phenolic Compounds (mg × 100 g of FW)	Canopy Zone	Treatments			
		Uncovered Trees		Covered Trees	
		K−	K+	K−	K+
Hydroxycinnamic acid 1 *	Lower	34.6 ± 4.5 Aba	26.8 ± 3.6 BCb	21.7 ± 5.2 Ca	39.4 ± 1.9 Aa
	Upper	27.5 ± 3.5 Ba	43.2 ± 1.3 Aa	29.1 ± 2.4 Ba	28.8 ± 5.6 Bb
Hydroxycinnamic acid 2 *	Lower	91.9 ± 20.1 Aa	85.6 ± 13.2 Aa	92.8 ± 26.0 Aa	123.7 ± 2.2 Aa
	Upper	56.9 ± 5.4 Ba	104.1 ± 9.9 Aa	93.6 ± 11.8 Aa	52.5 ± 9.7 Bb
Quinic acid derivative	Lower	5.1 ± 0.3 Bb	5.8 ± 0.2 Ab	5.4 ± 0.26 Ba	5.4 ± 0.1 Aa
	Upper	5.8 ± 0.1 Ba	6.4 ± 0.1 Aa	4.9 ± 0.1 Ca	5.4 ± 0.3 Ba
Total hydroxycinnamic acids	Lower	120.0 ± 24.6 Aa	118.2 ± 16.7 Aa	131.38 ± 31.0 Aa	167.6 ± 2.3 Aa
	Upper	90.18 ± 9 Ba	153.78 ± 10.7 Aa	127.73 ± 14.29 Aa	86.83 ± 15.1 Bb
Quercetin-rutinoside	Lower	7.3 ± 0.7 Ab	8.7 ± 0.4 Ab	8.5 ± 1.2 Aa	10.0 ± 0.9 Aa
	Upper	11.1 ± 1.4 Ba	15.8 ± 1.1 Aa	9.4 ± 0.9 Ba	9.7 ± 0.3 Ba
Flavonol 2 *	Lower	3.1 ± 0.2 Bb	3.9 ± 0.4 Aa	2.7 ± 0.1 Bb	4.0 ± 0.4 Aa
	Upper	4.9 ± 0.3 Aa	5.4 ± 0.6 Aa	4.2 ± 0.4 Ba	3.7 ± 0.3 Ba
Total Flavonols	Lower	10.5 ± 0.8 B	12.7 ± 0.8 Ab	11.12 ± 1.2 Bb	14.1 ± 1.3 Aa
	Upper	18.27 ± 2.48 A	21.19 ± 1.6 Aa	13.7 ± 1.3 Ba	13.4 ± 0.58 Ba
Cyanidin-3-rutinoside	Lower	74.0 ± 5.7 Ba	55.9 ± 5.2 BCa	39.9 ± 13.5 Cb	109.1 ± 18.5 Aa
	Upper	67.5 ± 5.5 Ca	204.5 ± 13.6 Ab	103.8 ± 15.7 Ba	125.8 ± 9.5 Ba
Peonidin-3-rutinoside	Lower	9.2 ± 0.8 Aa	5.1 ± 0.1 Bb	3.7 ± 0.9 Bb	9.4 ± 1.7 Aa
	Upper	6.1 ± 0.7 Ba	8.9 ± 0.5 Aa	6.9 ± 0.7 Ba	10.9 ± 1.3 Aa
Total Anthocyanins	Lower	83.3 ± 6.4 Ba	61.1 ± 5.3 BCb	43.7 ± 14.2 Cb	131.6 ± 17.5 Aa
	Upper	73.7 ± 6.2 Ca	213.7 ± 13.9 Aa	110.9 ± 16.3 Ba	136.9 ± 10.7 Ba

Means ± standard errors (SE) are shown. Statistically significant differences between treatments for each canopy zone are represented by different uppercase letters (horizontally), whereas differences between canopy zones for each K treatment are represented by vertical lowercase letters, based on the LSD Fisher multiple range test (*p* ≤ 0.05).

**Table 6 plants-12-00050-t006:** The statistical significance of each studied factor and its interaction with phenolic compounds detected in cherry fruits..

Phenolic Compounds	Zone	Significance		
		Significance	Cover	Potassium × Cover
Cyanidin-3-rutinoside	Lower	*	n.s.	***
	Upper	***	n.s.	***
Peonidin-3-rutinoside	Lower	n.s.	n.s.	***
	Upper	***	n.s.	n.s.
Total Anthocyanins	Lower	*	n.s.	***
	Upper	***	n.s.	***
Quercetin-rutinoside	Lower	n.s.	n.s.	n.s.
	Upper	*	***	*
Flavo(a)nol 2 *	Lower	***	n.s.	n.s.
	Upper	n.s.	**	n.s.
Total Flavo(a)nols	Lower	**	n.s.	n.s.
	Upper	n.s.	***	n.s.
Hydroxycinnamic acid 1 *	Lower	n.s.	n.s.	**
	Upper	*	n.s.	*
Hydroxycinnamic acid 2 *	Lower	n.s.	n.s.	n.s.
	Upper	n.s.	n.s.	***
Quinic acid derivative	Lower	*	n.s.	n.s.
	Upper	***	***	n.s.
Total hydroxycinnamic acids	Lower	n.s.	n.s.	n.s.
	Upper	n.s.	n.s.	***

Asterisks indicate a significant interaction between the factors (* *p* ≤ 0.05. ** *p* ≤ 0.01. *** *p* ≤ 0.005). n.s. indicates non-statistical significance.

**Table 7 plants-12-00050-t007:** The effect of two potassium spray treatments, conventional K regime (4 sprays during the season: K- treatment) or intensive K regime (7 sprays during the season: K+ treatment), on the organic acid composition of sweet cherry fruits (cv. Regina). Fruits produced in the lower and upper zone of the canopy were evaluated separately. Statistically significant differences between treatments are represented by different letters above bars, based on the LSD Fisher multiple range test (*p* ≤ 0.05). Error bars indicate the standard error for each dataset. Abbreviation: OA (organic acids).

Organic Acid (g × Kg^−1^ FW)	Treatment			
	K-		K+	
	Lower	Upper	Lower	Upper
Malic acid	0.99 ± 0.08 c	1.21 ± 0.014 ab	1.2 ± 0.015 ab	1.27 ± 0.12 a
Oxalic acid	0.11 ± 0.007 c	0.12 ± 0.007 c	0.16 ± 0.004 b	0.20 ± 0.027 a
Citric acid	0.136 ± 0.032 a	0.11 ± 0.019 ab	0.12 ± 0.002 ab	0.085 ± 0.012 b
Total OA	1.24 ± 0.11 c	1.4 ± 0.016 b	145.5 ± 15.6 b	156.3 ± 14.9 a

Means ± standard errors (SE) are shown. Statistically significant differences between treatments for each canopy zone are represented by different lowercase letters (horizontally), based on the LSD Fisher multiple range test (*p* ≤ 0.05).

## Data Availability

The data presented in this study are available in the article.

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
