# Peer review of "Plastic Covers and Potassium Pre-Harvest Sprays and Their Influence on Antioxidant Properties, Phenolic Profile, and Organic Acids Composition of Sweet Cherry Fruits Cultivated in Southern Chile"

_plants, 2022, doi:10.3390/plants12010050_

Round 1

Reviewer 1 Report (Previous Reviewer 3)

A scientific publication requires new insights which will be important for other researchers. This manuscript is not covering the biologic role of the treatments used as the Journal's aim require.

Author Response

Dear Reviewer,

This article presents important responses of sweet cherries fruit quality and composition submitted to different nutrients fertilizations. This gives an approach to improve fruit quality in an important sector of the industry of Chile, where we have some constraints factor like soil composition and climatic variables which are relevant for the growers in this location to be competitive in the export industry.
Previously, we published in this journal an article titled "Impact of Potassium Pre-Harvest Applications on Fruit Quality and condition of Sweet Cherry (Prunus Avium L.) cultivated under plastic cover in Southern Chile Orchards.", which is related to this article. This publication gives more progress in the action mechanism of the potassium in the increase of Quality at harvest and post harvest of sweet cherrie trees.

We hope to have answered your question.
Kind regards,

Reviewer 2 Report (Previous Reviewer 2)

I suggested a slight change in the title:
…Sweet cherry fruit cultured in in Southern Chile

Is the title that you have plastic cover in the second sentence that you have plastic covering, which one is proper?

Abstract

also increased by application                      also increased with the application

on maintaining

Introduction, results, discussion, conclusions

Suggestions for changes to the text were made in the pdf file attached to the review.

Materials and methods need to be improved Too many procedures are written in one string in the form of text. If possible, please list the individual analyses as well as the activities that were needed to perform these analyses. Please give a bigger breakdown into paragraphs and paragraphs.

The discussion in the work is too extensive in relation to the whole work. Very often in the discussion the results are repeated, numerical data are quoted, which have already been discussed or are in tables. This is unnecessary, moreover, the discussion also touches on aspects that have not been analyzed and are the background for research or have been studied by other researchers. In a discussion where there are few paragraphs difficult to read, as in materials and methods, a greater division should be introduced.

Author Response

Dear reviewer, 

Please see the attachment where you can find the aswer to your comments

Kind Regards

Reviewer 3 Report (Previous Reviewer 1)

I don't have no further comments.

Author Response

Dear reviewer, 

Thank you for your valuable review.

Kind Regards

Reviewer 4 Report (New Reviewer)

The manuscript entitled "Influence of Plastic Cover and Potassium Pre-harvest Sprays on Antioxidant Properties, Phenolic Profile and Organic Acids Composition on Fruits of Sweet Cherry Cultivated in Southern Chile”.

I think the quality of this work was good and can be accepted with some major revisions. First of all, it is necessary to revise English thought the manuscript which is full of naive mistakes that make very hard to understand the meaning of several sentences. Some examples:

Lines 48-50: The sentence is not clear to me. Southern Chile has or has not suitable climatic conditions for the cultivation of sweet cherry? Please, specify

Lines 55-57: Please rewrite as follows: The use of plastic covers, acting as a physical barrier that prevents direct contact of water with the cherry surface, has been reported to be an effective strategy to reduce rain-induced cracking in sweet cherry fruits

Lines 57-60: Please rewrite as follows: Nevertheless, several works showed that plastic covers can also negatively alter fruit quality such as firmness, resistance of fruits to mechanical damage during harvest, handling, and long post-harvest trips

Lines 71-73: Please rewrite as follows: Adequate K fertilization has demonstrated to increase fruit firmness, calibre and weight, soluble solids content and acidity in sweet cherries cultivated in Turkey [19].

Lines 73-77: The sentence is not clear. Please rewrite.

Lines 78-82: The sentence is not clear. There are some repetitions. Please rewrite.

Line 86: Please rewrite as follows: As far as we know, there are no reports regarding …

Lines 89-91: Sweet cherry, containing high levels of dietary fiber, carotenoids, ascorbic acid, anthocyanins and phenolic acids, is greatly valued by consumers not only due to its taste and color, but  also to its nutritional value and beneficial health effects [26,27].

Line 92: “consisting” should be: “consist”

Line 98: “due to the increase” should be: “as the increase”

Lines 164-166: The sentence is not clear. Please rewrite.

Lines 199-200: The sentence is not clear. Please rewrite.

Line 228: “are” should be: “is”

And so on

Author Response

Dear reviewer, 

Please see the attachment where you can find the aswer to your comments

Kind Regards

Round 2

Reviewer 1 Report (Previous Reviewer 3)

I will stand with my first opinion about the manuscript. A scientific publication requires new insights which will be important for other researchers. This manuscript is not covering the biologic role of the treatments used as the Journal's aim require. The manuscript is not suitable for publication in a scientific Journal with impact factor 4.7.

Reviewer 2 Report (Previous Reviewer 2)

Compared to the original manuscript, the current version has been significantly improved. My doubts of a methodical nature have been clarified in an acceptable way. Given the positive reviews of other reviewers and the comments they had previously received, which were taken into account by the authors, I am also inclined to accept the work.

Reviewer 4 Report (New Reviewer)

The manuscript has been improved and can be published in the present form.

This manuscript is a resubmission of an earlier submission. The following is a list of the peer review reports and author responses from that submission.

Round 1

Reviewer 1 Report

Manuscript 'Influence of Plastic Cover and Potassium Pre-harvest Sprays on Antioxidant Properties, Phenolic Profile and Organic Acids Composition on Fruits of Sweet Cherry Cultivated in Southern Chile' paper is well written, the text is clear and easy to read, and the statistical and other methods used are appropriate. This allows the authors to draw definitive conclusions.
The main question addressed by the research is whether we can overcome the negative effects of plastic covers on the quality of sweet cherries, i.e., lower organic acid content and fruit color, by using potassium. The topic is relevant and interesting, especially in areas where sweet cherry production is uneconomical without plastic covers.
The problem is new in sweet cherry, but has been tested in some other plants. Each species has its own specific way of responding to different stress factors. Thus, the cherry tree also responds in its own way to being covered with a plastic sheet. In situations where growing cherries without a cover is not economical, it is very important that you try to eliminate the negative effects of the cover through technical measures.
the conclusions are consistent with the evidence and arguments presented,and they addressed the main question posed.

Reviewer 2 Report

Review Plants 2001044

Influence of Plastic Cover and Potassium Pre-harvest Sprays on Antioxidant Properties, Phenolic Profile and Organic Acids Composition on Fruits of Sweet Cherry Cultivated in Southern         Chile

Intense rainfall during fruit ripening can not only hinder the organization of harvesting, but also cause cracking of fruits, which as a result of this process lose their commercial value.  In many countries, various types of research have been carried out to assess the methods of protecting fruits against cracking. One of the more effective methods is the use of a plastic film roof above the crowns, so that the water does not come into direct contact with the fruit. However, the cost of such orchard coverage is relatively large, and although cherries are a delicatessen fruit, the expenditures do not have to compensate for the costs.

L 19: I suggest to rewrite the first sentence

In rainy locations, sweet cherries are cultivated under plastic covers, preventing fruit cracking but decreasing cherry firmness and acidity. In rainy areas, sweet cherries are cultivated under plastic covers, preventing cracking of the fruit but decreasing cherry firmness and acidity of the cherries.

L 25: antioxidant capacity in fruits      newer in, should be ‘of fruits’

L 29: higher in fruit from covered trees  → in the fruit of covered

Keywords

Some of the keywords coincide with the words used in the title. Please verify this and supplement it with new words.

L46 and others: Remove nomenclature errors: variety →cultivar. Botanical nomenclature is regulated by the International Code of Nomenclature for Cultivated Plants (ICNCP).   The name cultivator refers to a botanical unit produced by human breeding methods, while a botanical unit that was created by way of individual development without human intervention and separated from a higher systematic unit, i.e. from a species, is called variety. Please check all the work and enter the name of this taxa correctly.

L48: for sweet cherry cultivation  → for the cultivation  of sweet cherries,

L54: Thus,…..in    Therefore,…..of

L57: Nonetheless,     Nevertheless

L74-76: I cannot agree with this statement. Of course I see that you provide some references but in my opinion  the key of consumer acceptance  is the size of fruits. For sure like this is in Europe but I think that as well on international markets the price of the sweet cherries is strongly dependent of the size of the  fruit. I don't see any literature review discussing this feature in your introduction.

L83:   Language                      of organic acids composition in fruits of …. of the composition of organic acids in fruits of …

L90: Language      In addition, sweet cherries are characterized to have anthocyanins as major phenolics, being 3-rutinoside and 3-….  →.  Furthermore, sweet cherries are characterized to have anthocyanins as the main phenolics, with 3-rutinoside and 3-

L98-100:  In relation to the content of the work, the goal is written poorly and imprecisely. Please edit it.

L 101-103: What is that? Template?

Results:

One of the most important elements that seems to have been raised only now in the results and discussion is the division of all observations depending on the place in the crown of the tree.  There is not a word of introduction about it and the discussion that suddenly arises on the subject makes it completely incomprehensible.

M&M

Reading your introduction, I got an impression that weather condition is a key factor which justify your experiment. Even if precipitation it's very high it doesn't mean that it can cause the cherry cracking. The danger of this process is only if you have precipitation shortly before or during ripening.  148 mm of precipitation is not rainy area. Please provide detailed weather data min. monthly, but better weekly.

What mean “the nearest weather station” 1km or 100 km?

I am impressed by the enormity of the conducted research, chemical analyzes, which could give very interesting answers to the question: how does covering the cherry orchard affect the basic quality characteristics of the fruit as well as the internal chemical quality expressed both by the oxydoreduction potential and the content of biologically active compounds.

The way the results are presented is also very interesting. Tables where many single-factor analyses are placed at the same time are transparent. They allow you to analyze the differences between two independent factors in one table.  In addition, the standard error and the row regarding the significance of differences make these tables very easy to read and the results appeal to the imagination. Similarly, PCA charts, where the correlation value is presented in an interesting way, are well thought out and well presented.

The idea of the experiment itself is also very good, but the experiment should either be carried out in a completely different area where building rain protection makes sense or it should be repeated again in years when rainfall is high. Precipitation at the level of 148 mm does not justify the construction over the orchard and covering the orchard with foil. Under the presented conditions, spraying outside the cover and under the cover with any substances and assessing the impact of this spraying does not make sense. There is no rainfall both under cover and outside the cover. In this way, the underlying factor has not really been studied. The expenses on chemical analyses is much greater than the purchase of a Meteo station that could measure the actual weather conditions at this point.  Collecting data using a station that is considered to be the closest and which, according to Google maps, is more than 200 km away, in this case is completely pointless. Given these elements, I am forced to reject the work in its entirety. I hope that you will be able to carry out this experiment again in conditions that can be considered objective.

If you plan repeat an experiment please use as a material another cultivar proneto cracing more than Regina.

Reviewer 3 Report

In this work, entitle "Influence of Plastic Cover and Potassium Pre-harvest Sprays on Antioxidant Properties, Phenolic Profile and Organic Acids Composition on Fruits of Sweet Cherry Cultivated in Southern Chile", the authors report the role of Plastic Cover and Potassium treatments on sweet cherry quality traits.

Overall, this works could offer new insights that are important for researchers. However, the authors did not found any specific pathway to be induced by pre-harvest K foliar application and/or plastic covering, despite the soundness of the experimental design.

The references used are too many (108!), however the authors should provide more information in the discussion section and not just reporting the results of the current research.